

# Global distribution of nearshore slopes with implications for coastal retreat

Panagiotis Athanasiou [1,2], Ap van Dongeren [1,5], Alessio Giardino [1], Michalis Vousdoukas [3,4], Sandra Gaytan-Aguilar [1], Roshanka Ranasinghe [5,2, 1]

[1] Deltares, Delft, The Netherlands
[2] Water Engineering and Management, Faculty of Engineering Technology, University of Twente, Enschede, The Netherlands
[3] European Commission, Joint European Research Centre (JRC), Ispra, Italy
[4] Department of Marine Sciences, University of the Aegean, Mitilene, Greece
[5] IHE Delft Institute for Water Education, Delft, The Netherlands

*Correspondence to*: Panagiotis Athanasiou (Panos.Athanasiou@deltares.nl)

**Abstract**

Nearshore slope, defined as the cross-shore gradient of the subaqueous profile, is an important input parameter which affects hydrodynamic and morphological coastal processes. It is used in both local and large-scale coastal investigations. However, due to unavailability of data, most studies, especially those that focus on continental or global scales, have historically adopted a uniform nearshore slope. This simplifying assumption could however have far reaching implications for predictions/projections thus obtained. Here, we present the first global dataset of nearshore slopes with a resolution of 1 km at almost 620,000 points along the global coastline. To this end, coastal profiles were constructed using global topo-bathymetric datasets. The results show that the nearshore slopes vary substantially around the world. An assessment of sea level rise (SLR) driven coastline recession (for an arbitrary 0.5 m SLR) with a globally uniform coastal slope of 1:100, as done in previous studies, and with the spatially variable coastal slopes computed herein shows that, on average, the former approach would under-estimate coastline recession by about 40%, albeit with significant spatial variation. The final dataset has been made publicly available at https://doi.org/10.4121/uuid:a8297dcd-c34e-4e6d-bf66-9fb8913d983d.

## 1 Introduction

Ten percent of the world's population lives in low-lying coastal areas;  i.e. less than 10 m above the current mean sea level (MSL) (McGranahan et al., 2007). In the future, the population density in these areas is expected to increase even more due to high rates of population growth and urbanization (Neumann et al., 2015). At the same time, coastal areas are exposed to a number of marine hazards that can lead to flooding or erosion. As the coastline comprises various different landforms, the response to these hazards can vary significantly both spatially and temporally. One of the most vulnerable coastal types are sandy coasts which are highly dynamic and can change in response to extreme marine events (McCall et al., 2010),  long



term trends in MSL (Zhang et al., 2004), natural gradients in alongshore sediment transport (Antolínez et al., 2018) or human interventions (Giardino et al., 2018a; Luijendijk et al., 2018; Mentaschi et al., 2018).

Local, regional or global studies that seek to quantify natural or human induced coastal change require high quality nearshore bathymetry and sub-aerial topography data. However, as such data are rarely available, previous studies, especially at the regional or global scale, have had to rely on a number of limiting assumptions, such as globally uniform nearshore or beach slopes (Hinkel et al., 2013; Melet et al., 2018). The nearshore area is here defined as the part of the cross-shore profile between the depth of closure $d_c$ (i.e. the offshore limit where sediment transport is zero) and the shoreline (MSL). The nearshore slope (the ratio of the $d_c$ over the horizontal distance between the $d_c$ and the shoreline) modulates the wave transformation (Battjes, 1974) and the total water levels (Serafin et al., 2019), while it is associated with geomorphological processes at various temporal scales (Bruun, 1962; Wright and Short, 1984; Dean, 1991). Therefore, a global assumption of uniform slope likely hides the spatial variability of coastal hydro- and morphodynamics.

Accurate information on the offshore limit of the nearshore area (i.e. the depth of closure $d_c$) and the nearshore slope is important for numerous coastal engineering applications. For example, when behaviour-oriented models are employed, the depth of closure and the nearshore slope are crucial model inputs (Larson et al., 2004; Ruggiero et al., 2010; Hanson, 2014). At data poor locations, when bathymetric data are not available, an equilibrium bathymetric profile (Dean, 1991) is often assumed and the local $d_c$ is needed to define the offshore profile limit (Udo and Takeda, 2017). Various assessments of future coastal recession due to sea level rise (SLR) have used the Bruun rule (Bruun, 1962) to quantify coastal retreat (Zhang et al., 2004; Hinkel et al., 2013; Baron et al., 2014; Monioudi et al., 2017; Udo and Takeda, 2017; Ballesteros et al., 2018; Giardino et al., 2018b), an approach which is very sensitive to the nearshore slope. Furthermore, several studies, both at the global scale (Hinkel et al., 2013) and regional scale (Brutsché et al., 2016; Toimil et al., 2017), have estimated the $d_c$ using empirical formulae (Hallermeier, 1978; Birkemeier, 1985; Nicholls et al., 1998) that relate $d_c$ to wave parameters.

Against the foregoing background, the present study sets out to provide the first ever global database of nearshore slopes and their associated depths of closure, which can be used in local, regional or global studies which aim to quantify coastal response to natural (i.e. marine extremes and climate change) or human induced phenomena. Our methods utilize available open-source global datasets of topography, bathymetry and shoreline location in order to create a seamless representation of coastal morphology. Furthermore, global wave statistics are used to estimate $d_c$. The resulting dataset consists of an estimation of the local nearshore slopes at ~620.000 points along the global coastline.

This paper is structured as follows: Section 2 describes the methods used for the creation of the seamless coastal morphology map, the calculation of the global depths of closure, the definition of the coastal profiles and the calculation of the nearshore slopes. Section 3 presents the results of the analysis, consisting of a global dataset of depths of closure and nearshore slopes while Sect. 4 compares the results versus observed data at several locations around the world. The paper concludes with a



discussion of key points, including a global analysis of the sensitivity of coastal recession estimates using the Bruun rule to the nearshore slopes, and the main conclusions.

## 2    Materials and methods

### 2.1    Workflow to obtain nearshore slopes

The estimation of nearshore slope required the following steps (Fig. 1). First, a seamless topo-bathymetric map was created (Sect. 2.2.1), then cross-shore transects were drawn with an accompanying elevation profile (Sect. 2.2.2). Using global wave data the depths of closure were determined (Sect. 2.3), after which the nearshore slopes could be determined (Sect. 2.4).

### 2.2    Coastal profiles

#### 2.2.1    Global topo-bathymetric data

A cross-shore contiguous representation of the coastal profile is needed in order to accurately derive geometrical parameters such as the nearshore slope. Presently, elevation data are separated between topographic, i.e. representing the subaerial surface of the Earth, and bathymetric, i.e. representing the subaqueous seabed. Here we merge two of these datasets in order to create a seamless elevation layer with a smooth transition between land and water.

The topographic layer that was used is the MERIT digital elevation model (DEM) (Yamazaki et al., 2017), which is an
improved version of the existing space borne DEMs SRTM3 v2.1 (Farr et al., 2007) and AW3D-30m v1 (Tadono et al., 2015), after removing multiple error components (i.e. absolute bias, stripe noise, speckle noise, and tree height bias). It represents the terrain elevations at a 3 arc-sec resolution (~90 m at the equator) and covers land areas between 90N-60S, referenced to EGM96 geoid. The bathymetric layer was obtained from the General Bathymetric Chart of the Oceans (https://www.gebco.net) which is based on ship track soundings with interpolation between soundings guided by satellite-
derived gravity data. The version used was the GEBCO_2014 Grid (Weatherall et al., 2015), which has a 30 arc-sec resolution (~900 m at the equator) and it is generated by the aggregation of heterogeneous data types assuming all of them to be referred to MSL.

As the two datasets might overlap, the Open Street Map (OSM) (OpenStreetMap contributors, 2015) coastline was used to create a mask for land and water. For the final elevation dataset, a common grid was used, with a resolution corresponding to
the higher resolution of the MERIT DEM dataset. Subsequently, the bathymetric data were interpolated only at the cells seaward of the OSM coastline. This was performed by a linear interpolation of the values at neighbouring grid points in the $x$ and $y$ dimension. One of the key elements when merging bathymetry and elevation data is to create a smooth transition between land and water. We have applied a Low Pass Filter to decrease disparities between pixel values by averaging nearby pixels. The smoothing technique was applied only on the cells along the coastline. For each cell the vertical offset $\Delta z$ was



computed (as the difference in elevation with respect to its neighbouring cell). For the smoothing procedure different windows (i.e. number of grid cells around the grid cell to be smoothed) were used according to $\Delta z$. Smoothing window sizes of 3x3 were applied at areas where $\Delta z$ was up to 10 m, 5x5 grid cells where $\Delta z$ higher than 10m and 9X9 is few geographic areas where topography data present a big discrepancy with respect to the bathymetry.

### 2.2.2    Global elevation transects

The global coastline was defined using the OSM dataset of 2016. Since the full scale coastline was found to be too detailed for the purposes of the present work, the level 8 generalized version of the dataset was used, which is a smoothed version that removes small details and has been previously used in global coastal assessments (Luijendijk et al., 2018).  About 1,000,000 cross-shore transects were defined along the OSM coastline, spaced at 1 km intervals. The middle point of each
transect was defined at the OSM location, with a 4km extension both to the landward and seaward direction. The profile length was chosen after testing different lengths to ensure that the active coastal profile is captured while minimizing data storage. The 1 km spacing was chosen in order to have a good coverage of the alongshore variability of the coastal profiles, while keeping the computational costs at a feasible level. Along each transect, a set of equidistant points was defined at an interval of 25 m, while flagging the locations where the elevation values were extracted from the global grid.

The global transects (~1,000,000) were first filtered to exclude transects located along coasts covered with ice, river inlets, ports and other significant human interventions, using the global transects of Luijendijk et al. (2018). This reduced the total number of transects to about 780,000 transects. Additionally, and specifically for the application described in Sect. 4.2, the sandy beach location data from Luijendijk et al. (2018) were used in order to identify the transects that were sandy, resulting in about 215,000 sandy transects. Note that gravel beaches are included in this dataset and are herein refered to as sandy.

## 2.3    Calculation of the depth-of-closure

The depth of closure was calculated using the formulation of Nicholls et al. (1998):

$$d_c = 2.28 H_{e,t} - 68.5 \left( \frac{H_{e,t}^2}{g T_{e,t}^2} \right), \tag{1}$$

where $H_{e,t}$ is the significant wave height that is exceeded only 12 hours per $t$ years, $T_{e,t}$ is the associated wave period, and $g$ is the gravitational acceleration. The selection of the length of the time series $t$ in this formula is directly associated with the
temporal scale of the application of the depth of closure. This means that, for example, applications that consider temporal scales in the order of decades (i.e. coastline models, Bruun rule) should use a $t$ that is consistent with the assessment scale. Naturally, the temporal extent of the time series is dictated by the length of the available observations or modelling results. For example, Udo and Takeda (2017) applied Eq. (1) for SLR induced coastal recession assessments along the Japanese coast taking into account the effects of time scale for their 100-year Bruun rule application by using the maximum wave
heights in their 5 years long wave record.



Here we applied Eq. (1) using the full time series of significant wave heights $H_s$ and peak wave periods $T_p$ from the global reanalysis covering the period between 1980 and 2014 presented by Vousdoukas et al. (2018b). The wave parameters were available every 3 hours at offshore locations covering the ice-free coasts.

## 2.4    Estimation of the nearshore slope

For each of the transects, the cross-shore location of the depth of closure $d_c$ was identified using the elevation profile and the $d_c$ value derived as described in this section. The location of the depth of closure was calculated by finding the first (i.e. moving from the most offshore point towards land) point which was shallower than the $d_c$ and then linearly interpolating in between the adjacent cross-shore points along the transect. Then the cross-shore location of the shoreline (MSL) point was estimated using a similar method. In case of multiple shoreline points (e.g. barrier islands) the point that was located closer to OSM defined point was chosen. The cross-shore distance between these two points was used as the horizontal length of the nearshore area $L$. The nearshore slope $\tan(\beta_{ns})$ was then calculated as the ratio $\frac{d_c}{L}$ .

At locations in which the calculation could not be carried out as described above, an error code was assigned to describe the cause of the calculation failure (Fig. S1-S2). For example, as the OSM coastline does not represent the open coast only, but includes a substantial number of areas with elevation that is far from the conventional profile presented in Fig. 1, the calculation of the slope in this way was not always successful. This was typically the case for coastlines dominated by fjord features (e.g. Norway, Chile). Furthermore, some transects did not have any offshore point in close proximity for the calculation of the depth of closure. The transects that the nearshore slope could not be calculated were about 160,000 of the total 780,000.

At some other locations (mostly embayed or protected areas) the depth of closure point was located seaward of the most offshore profile point. Therefore, the deepest point was taken as the depth of closure $d_c$. Additionally, due to the merging process of the bathymetric and topographic elevation data, elevation steps close to the shoreline were observed in some transects. These transects were determined by identifying abrupt changes of the slope close to the shoreline. A warning code was also assigned to such transects (Fig. S1-S2).

## 3    Results

### 3.1    Global depths of closure

The resulting global distribution of $d_c$ (Fig. 2) showed that on open ocean coasts, $d_c$ generally increased with latitude because of the wave height, while $d_c$ was smaller along the shores of more wave-sheltered seas. The global average value of $d_c$ was approximately 13 m, while it reached values of more than 20 m at some areas at higher latitudes (Fig. 3).





### 3.2 Nearshore slopes

The calculated nearshore slopes (Athanasiou, 2019) showed high spatial variability globally (Fig. 4 with continental zooms in Fig. S3-S8). The histogram of the calculated nearshore slopes (Fig. 5a) revealed that the most commonly computed nearshore slopes were around 0.005. Additionally, there was a high probability of occurrence of nearshore slopes larger than
0.2 due to the high number of transects at fjords and other steep coastlines. Furthermore, we detected several mildly sloping locations (i.e. nearshore slope lower than 0.002). Such mildly sloping coastlines are mostly characterized by high tidal range and/or muddy coastlines (e.g, Northwest Australia). The median nearshore slope worldwide was 0.007, while the 10$^{th}$ and 90$^{th}$ percentiles were 0.001 and 0.565 respectively (Fig. 5b). When only the sandy transects were considered (see Sect. 2.2.2), the most common nearshore slope shifted to 0.01, which corresponds with the globally uniform profile slope assumed
in previous global scale studies (Hinkel et al., 2013).

The nearshore slope value is dependent on various geological and hydrodynamic factors, such as wave characteristics, sediment size, sediment supply and large scale geological and tectonic processes, among others. For that reason we aggregated the nearshore slopes along various regions around the world, defined heuristically considering geographical proximity and their corresponding oceans or seas. A distinct variation of the nearshore slopes can be seen among regions
(Fig. 6). The three regions with the on average steeper slopes were the Pacific Islands, the Norwegian Sea and the Mediterranean Sea with median values of 0.025, 0.020 and 0.014 respectively. On the other hand, the three regions with the mildest nearshore slopes were North Australia, East U.S and Gulf of Mexico and Southeast Asia with median values of 0.001, 0.002 and 0.003 respectively. The variance of the aggregated nearshore slopes per region (defined as the logarithmic difference between the 5$^{th}$ and 95$^{th}$ percentile) varied, with West S.America and North Australia showing the highest and
lowest variance respectively.

## 4 Comparison with local observations

### 4.1 Depth of closure

Different methods exist to estimate the $d_c$ and therefore the validation of this parameter can be quite challenging at a global scale. Nevertheless, we used a study at the U.S coast (Brutsché et al., 2016), which employed various formulas to calculate
$d_c$, including Eq.(1), to compare our results. We compared the $d_c$ at the offshore locations of the U.S. coast (Fig. 2) with the $d_c$ of the closest point from Brutsché et al. (2016), calculated using the same formula (Eq.(1)). Our predicted $d_c$ showed skill in capturing the spatial variation as calculated by Brutsché et al. (2016) (Fig. S9), but presented a positive bias of 3.34 m (Fig. S10). This can be attributed to the different location and temporal scale of the wave statistics (i.e. Brutsché et al. (2016) used nearshore transformed wave time series of a 20 years hindcast).



## 4.2    Nearshore slopes

The computed slopes were validated both qualitatively and quantitatively. The former was performed regionally using coastal classifications of the European and the Mediterranean coastlines, while the latter employed observations at sandy beaches derived from local surveys at 8 study sites around the world.

### 4.2.1    Qualitative validation

Two coastal classification datasets were used to inspect the computed nearshore slopes grouped according to geomorphological coastal types. One was the Mediterranean Coastal Database (MCD) (Wolff et al., 2018), which classifies the Mediterranean coastline in four classes: 1) sandy beaches, 2) unerodible coasts, 3) muddy coastlines and 4) rocky coasts with pocket beaches. The other was the EUROSION geomorphological classification (EUROSION, 2004) of the counties of the European Union (EU), which includes 20 different coastal types. The latter was reclassified to the four classes of the MCD database in order to reduce the total number of classes (Fig. S11)

The calculated nearshore slopes followed the expected variability between the different coastal typologies for both classification datasets (Fig. 7). For example, muddy coastlines had relatively milder nearshore slopes (median values of 0.0024 and 0.0025 for MCD and EUROSION datasets respectively). Sandy coastlines had a median nearshore slope of 0.0107 and 0.0066 respectively for the MCD and EUROSION datasets, which relates to the steeper sandy beach slopes observed along the Mediterranean coastline. Unerodible and rocky coastlines had on average relatively steeper slopes.

### 4.2.2    Quantitative validation

The study sites for the quantitative validation were selected according to data availability and a criterion of having at least 20 transects overlapping with the derived global dataset to enable the computation of error statistics per site. The validation focussed only on sandy coastline transects that are not adjacent to morphologically complex areas (e.g. inlets, river mouths) determined using satellite imagery. The ground truth sites (Fig. 8; Table 1) represent different coastal geomorphologies with varying hydrodynamic forcing and span across different continents. We performed the validation on the basis of the nearshore slopes, using the $d_c$ that were derived from the global analysis of wave data described in Sect. 2.2.2, for both local and global nearshore slope estimation. The local nearshore slope was calculated using the methodology described in Sect. 2.3.

When the elevation was represented by locally defined profiles, the comparison was performed using the closest globally derived transect. When point cloud data were available, they were first interpolated on a common grid and then the elevation was extracted along the global transects. The same process was performed when an elevation raster was available.

For a quantitative evaluation of the accuracy of the globally predicted slope values against the locally observed ones, various statistical parameters were calculated per site, as well as globally. This included the coefficient of determination $R^2$, the



normalized mean bias (NMB) and the normalized root mean square error (NRMSE). Note that there is no temporal coherence between the predicted and observed data, since the global elevation dataset is a merged product from temporally varying bathymetric sources (see Sect. 2.2.1) and the observed data were taken at different times. Therefore, it is expected that part of the calculated errors is due to potential temporal differences of the estimated and observed elevation data.

5 A regression analysis performed over all of the available global observations resulted in a normalized mean bias of almost - 25% and an $R^2$ value of 0.44 (Fig. 9) implying that that extracted nearshore slopes captured the spatial variability of the ground truth data. The negative overall bias indicated that the calculated global slopes were on average milder than those observed.

The error statistics per validation site (Table 2) showed that overall, $R^2$ values were above 0.6 except for North California, 10 Recife and South California where poorer agreement was found (i.e. 0.38, 0.25 and 0.3 respectively). At all validation sites (except Monterey Bay) a negative bias was observed. The NRMSE varied per study site, with values lower than 35% for five out of the eight validation sites. The high NRMSE at Emilia Romagna can be attributed to the quite high bias, but nevertheless the spatial variability was captured correctly ($R^2 = 0.7$). It should be noted that the number of overlapping sandy transects differed significantly between sites as the spatial extent of the available local surveys differed. For example, 15 at Emilia Romagna and Monterey Bay there were only 25 and 27 overlapping transects, while at South California 382.

## 5 Discussion

### 5.1 Spatial variation of nearshore slopes

As discussed in the previous section the results of the present study revealed a quite pronounced spatial variation in nearshore slopes worldwide (Fig. 4). For example, the West coast of North and South America had on average steeper 20 slopes in comparison to the East, which can be attributed to the swell dominated wave climate along the former (the same holds for the West and East coasts of Africa). The classification of the coasts with respect to the plate tectonics (Inman and Nordstrom, 1971) seems to be relevant to nearshore slopes distribution as well. Trailing edge coast with wider continental shelf width (e.g. North Sea, East Americas, North Australia) appeared to have milder nearshore slopes than leading edge coasts (e.g. West Americas and the biggest part of the Mediterranean). The coasts of the Gulf of Mexico, South and 25 Southeast Asia showed on average mild nearshore slopes, which can be attributed to the large (historic) sediment supplies by some major rivers therein.

### 5.2 Improvements in estimations of SLR-induced retreat of sandy coasts

Similarly to the overall data set (including non-sandy coasts), the subset of nearshore slopes of sandy coasts displays a considerable spatial variation, even though the commonly-used 1:100 nearshore slope assumption (Hinkel et al., 2013) lies in 30 the most probable range (Fig. 5). Using the Bruun rule (Bruun, 1962), we investigated the effect of applying the spatial



variation of the nearshore slope on the estimation of shoreline retreat due to sea level rise, in comparison with applying a uniform 1:100 slope as commonly done for sandy beaches.

The Bruun rule is a simple two-dimensional mass conservation principle, which predicts a landwards and upwards response of sandy profiles as a response to SLR. It is expressed as (Bruun, 1962):

$$R = \frac{L}{d_c} S \, , \tag{2}$$

Where $R$ is the horizontal coastal recession, $d_c$ is the depth of closure, $L$ is the horizontal length from the shoreline to the DoC and $S$ is the SLR. While used extensively for lack of an efficient better method, its use has been a controversial matter in literature during the last years (Cooper and Pilkey, 2004; Ranasinghe and Stive, 2009), as it is based on a number of assumptions and simplifications (i.e. equilibrium profile existence, cross-shore sediment balance). Nevertheless it is the only method that can deal with coastal recession due to SLR at large scale in a computationally feasible manner. Since $\tan(\beta_{ns}) = \frac{d_c}{L}$ this makes Eq.(2):

$$R = \frac{1}{\tan(\beta_{ns})} S \, , \tag{3}$$

Physically, there are limits of the nearshore slopes that can be encountered at sandy beaches related to the angle of repose of granular materials. For our dataset, the sandy transects (Luijendijk et al., 2018) were found to lie beyond these ranges in some cases. This can be attributed to local effects of geology, sediment budget, but also to data and methodological artefacts. Therefore, we applied upper and lower limits of 0.2 and 0.001 for the sandy nearshore slopes and constrained the slopes that lie beyond these limits to the limit values.

Following the linear relationship between the coastal recession $R$ and the nearshore slope in Eq. (3), the differences in the coastal recession under a given SLR scenario can show significant spatial variations. In order to highlight this, the ratio between the coastal recession $R_{calc}$ (i.e. using the nearshore slopes presented herein) and $R_{100}$ (i.e. using a 1/100 nearshore slope globally) was estimated globally for all sandy transects (Fig. 10). The results indicate that for a large number of areas around the world the assumption of 1:100 nearshore slopes underestimates the potential coastal recession with a median $\frac{R_{calc}}{R_{100}}$ of 1.39. However, it should be noted that there are also several locations with steeper slopes, where the 1:100 assumption overestimates the SLR induced erosion.

In order to highlight the potential impact of this effect, a coastal recession estimate was computed for all transects of each region (see Fig. 6) using Eq. (3) for an arbitrary SLR of 0.5 m across the global coastline. The median value of the coastal recession of all transects per region was calculated as a robust description of its central tendency (Fig. 11) and compared with the value of 50 m (which is the recession that would be calculated for a SLR of 0.5 m using the 1:100 slope assumption). The results show that, in most of the defined regions the assumption of a 1:100 nearshore slope would result in



the underestimation of coastal recession, especially in North Australia, South and Southeast Asia and the East U.S and Gulf of Mexico regions. However in some of the regions, such as the Mediterranean and the Norwegian Seas, and the West Americas the 1:100 slope assumption would lead to an overestimation due to the steeper nearshore slopes encountered on average there.

**5.3    Limitations**

The global scale of the present work introduced inevitable limitations related to data availability, computational efficiency and methodological constraints. Merging bathymetric and topographic data at a global scale can be quite challenging. This is associated with differences in spatial resolution, potential overlaps or differences in the vertical datum. Here we tried to resolve these issues by using the OSM coastline as the MSL to differentiate between sea and land and connect the elevation

data. Nevertheless, GEBCO is a combination of bathymetric data derived from variable sources and different techniques and in some cases might have differences in the vertical datum (Weatherall et al., 2015). This might introduce regional differences on the trustworthiness of the results. Furthermore, we expect that at locations where the nearshore slopes are steep, the resolution of the bathymetry is not high enough to capture a large number of bathymetry points between the shoreline and the depth of closure, hiding the details of the profile.  We believe that when bathymetrical dataset with better

resolution and accuracy become available in the future, the presented technique can be applied to update the global nearshore slopes estimations.

The wave statistics that were used to determine the depth of closure across the global transects of the present study were taken from the closest offshore point of the wave reanalysis output (see Sect. 2.3). Ideally, the wave time series of the 28 years used here should be transformed to nearshore conditions at a location just offshore of the depth of closure using Snell's

Law. This would require an assumption of a nearshore slope, then the calculation of $d_c$ and then the re-calculation of the nearshore slope using an iterative approach until the slopes converge. This was deemed to be outside the scope of this study, considering the large number of transects. Nevertheless, we found that the nearshore slopes were not very sensitive to small changes in the depth of closure.

With respect to the comparison of the dataset compiled in the present study with available observations (Sect. 4), it should be

mentioned that due to the scale of the study, it is quite challenging to perform a direct validation with the same spatiotemporal conditions. For example the comparison of the estimated $d_c$ was performed against the results of a study that used the same formula, but employing wave statistics with different temporal extent and at different locations (Sect. 4.1). Additionally, the local surveys used for the quantitative comparison of the nearshore slopes had various sources, used different measuring techniques and were collected at different times (Sect. 4.2.2). Naturally, it is expected that these

spatiotemporal differences had an impact on the error statistics computed herein.





In this study the nearshore slope was defined from the depth of closure until the shoreline. This could be a deviation from certain Bruun rule implementations that define the active profile slope from the depth of closure until the dune or berm crest (Zhang et al., 2004; Hinkel et al., 2013; Toimil et al., 2017; Udo and Takeda, 2017). Given that the available global topographic datasets lack the resolution to resolve dunes or berms (Vousdoukas et al., 2018c), their inclusion in the analysis

was not feasible, potentially resulting in an underestimation of the active profile slope (from the depth of closure to the top of the dune) along beaches with prominent dune systems. Nevertheless, since our dataset includes the nearshore slope and the depth of closure, one could calculate the slope from depth of closure to the dune or berm height if this information is available locally.

## 6    Data Availability

The final output data provide the location of each of the ~780,000 points along the global coastline together with: a) the depth of closure $d_c$, b) the nearshore slope $\tan(\beta_{sf})$ and c) an error /warning code in case the slope was not calculated or limitations in the profile were spotted. These data are given as a comma separated value file. The data can be downloaded via https://doi.org/10.4121/uuid:a8297dcd-c34e-4e6d-bf66-9fb8913d983d (Athanasiou, 2019).

## 7    Conclusions

A dataset of the global distribution of nearshore slopes at an alongshore resolution of 1 km is presented, using global elevation datasets and wave statistics. Depths of closure $d_c$ were estimated worldwide using an empirical formulation and long-term wave statistics derived from 34-years wave reanalysis. The average $d_c$ globally was almost 13m, attaining larger values at higher latitudes. The global median nearshore slope was 0.007, while most values were in the range 0.001 to 0.565 (10[th] and 90[th] percentile respectively). The most commonly computed nearshore slope was about 0.005, which increased to

0.01 when only sandy coasts were considered. However, the computed nearshore slopes exhibited substantial spatial variability, potentially corresponding with spatial variations in hydrodynamic forcing and geological conditions.

The dataset captured expected qualitative nearshore slope patterns when compared to available coastal classification datasets. Additionally, it captured the spatial variability of the nearshore slopes of sandy beaches at 8 validation sites from around the world, but with a negative bias of 25% (milder slopes than the observed ones).

The assessment of SLR driven coastline recession (for an arbitrary 0.5 m SLR) with globally uniform coastal slopes, as done in many previous studies, and the spatially variable nearshore slopes computed herein showed large differences between the recession amounts projected by the two approaches. Worldwide, the median coastline recession calculated with the nearshore slopes computed here was almost 40% larger than that computed by assuming a globally uniform 1:100 coastal slope, with the ratio between the two estimates varying substantially around the world. We believe that this dataset is a first

step towards capturing the spatial variability of coastal profile characteristics and enabling a correct spatial representation of coastal impacts.

**Author contribution**

PA, AvD, AG, MV and RR designed the study. PA carried out the data processing and analysis. MV provided the offshore
wave statistics. SGA composed the merged topobathymetric dataset. PA prepared the manuscript with contributions from all co-authors.

**Competing interests**

The authors declare that they have no conflict of interest.

**Acknowledgements**

This work has received funding from the EU Horizon 2020 Program for Research & Innovation, under grant agreement no 776613 (EUCP: "EUropean Climate Prediction system"; https://www.eucp-project.eu). Ap van Dongeren was funded in part by the Deltares Strategic Research Programme "Quantifying Flood Hazards and Impacts" while Alessio Giardino from the Research Programme "Flood Risk Strategies". Roshanka Ranasinghe is supported by the AXA Research fund and the Deltares Strategic Research Programme "Coastal and Offshore Engineering". We would like to acknowledge Arjen
Luijendijk for providing the location of the sandy transects from their study. We would also like to acknowledge the map data copyrighted OpenStreetMap contributors for using the map data available from https://www.openstreetmap.org.





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



Table 1: Validation case studies description

| Location (Country) | Data format | Resolution | Year | Source |
|---|---|---|---|---|
| Dutch Coast (The Netherlands) | Elevation profiles | Alongshore intervals of 250 m and cross-shore resolution of 5-10 m | 2011 | (Rijkswaterstaat, 2018) |
| Emilia Romagna (Italy) | Point cloud profiles | Alongshore intervals of ~500 m | 2012 | Topo-bathymetric surveys of beach profiles performed by Arpae-SIMC in the context of the Regional Topo-bathymetric Network |
| North California (U.S.) | Elevation profiles | Alongshore intervals of 100-200 m and cross-shore resolution of 2 m | Various (1929-2017) | (Barnard et al., 2014) |
| South California (U.S.) | Elevation profiles | Alongshore intervals of 100-200 m and cross-shore resolution of 2 m | Various (1930-2014) | (Barnard et al., 2014) |
| Monterey (U.S.) | Point cloud profiles | Alongshore intervals of 50-250 m | 2017 | (Stevens et al., 2017) |
| Long Island (U.S.) | Elevation raster | 1/9 arc-sec cell size | 2012 | https://www.ngdc.noaa.gov/mgg/inundation/sandy/sandy_geoc.html |
| New Jersey (U.S.) | Elevation raster | 10 m cell size | Various (1934-2013) | (Andrews et al., 2015) |
| Recife (Brazil) | Point cloud profiles | Alongshore intervals of ~150 m | Various (2012-2015) | (Vousdoukas et al., 2018a) |




Table 2: Error statistics per validation site on the basis of the nearshore slopes $tan(\beta_{ns})$.

| Validation Site | Transects | Mean observed nearshore slope | NMB (%) | NRMSE (%) | $R^2$ |
|---|---|---|---|---|---|
| Dutch Coast | 83 | 0.0083 | -18.0 | 24.4 | 0.62 |
| Emilia Romagna | 25 | 0.0062 | -62.6 | 130.4 | 0.70 |
| Long Island | 139 | 0.0092 | -31.4 | 33.3 | 0.80 |
| Monterey Bay | 27 | 0.0113 | 10.0 | 20.6 | 0.68 |
| New Jersey | 148 | 0.0100 | -53.8 | 40.1 | 0.87 |
| North California | 71 | 0.0156 | -11.2 | 18.4 | 0.38 |
| Recife | 62 | 0.0108 | -83.6 | 65.0 | 0.25 |
| South California | 382 | 0.0204 | -35.7 | 14.8 | 0.30 |



**Figures**

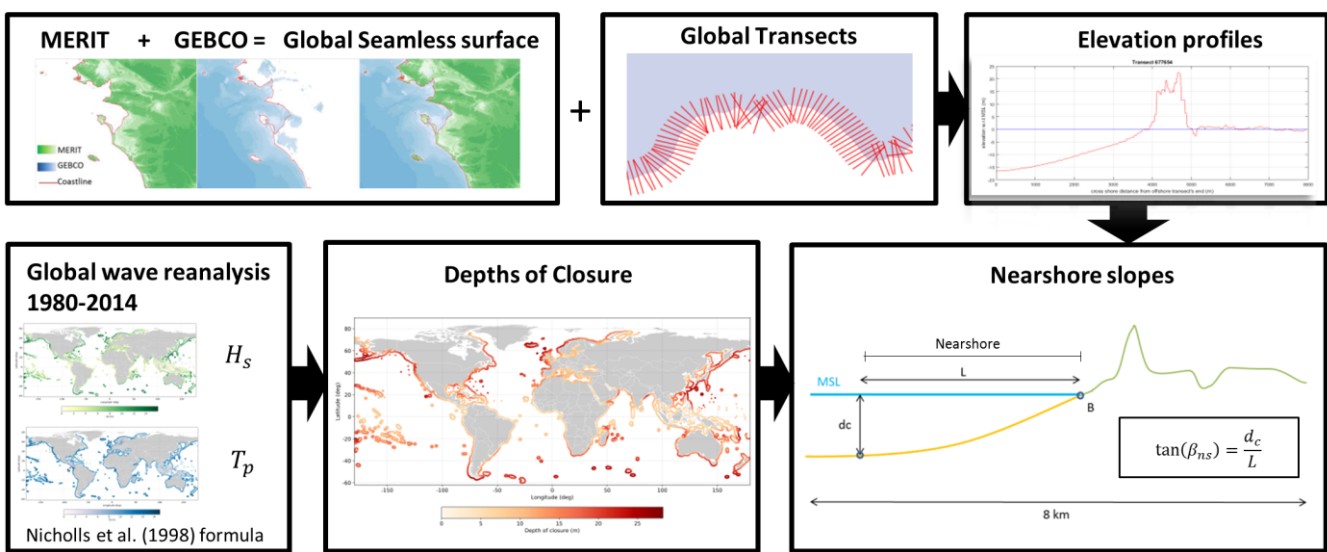

Figure 1: Work flow followed for the calculation of the nearshore slopes worldwide.



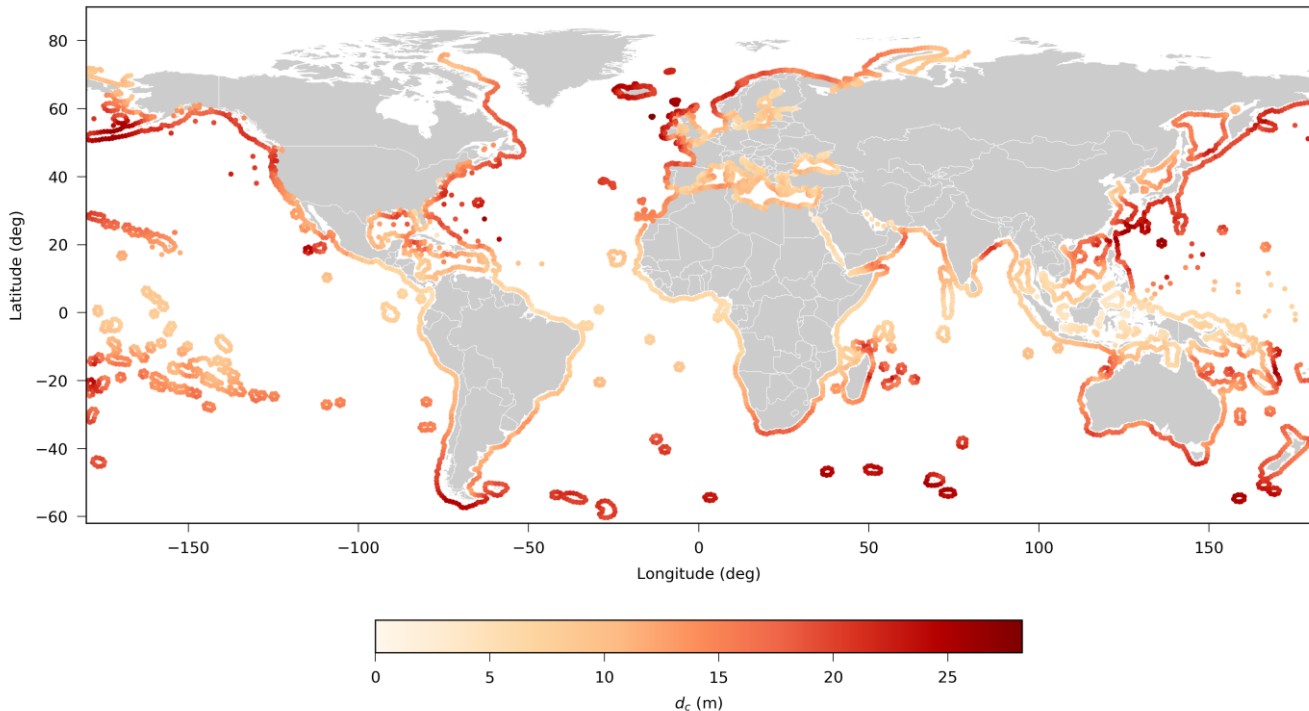

Figure 2: Depths of closure $d_c$ along the global coastline using the formulation by Nicholls et al., (1998) (Eq. 1).



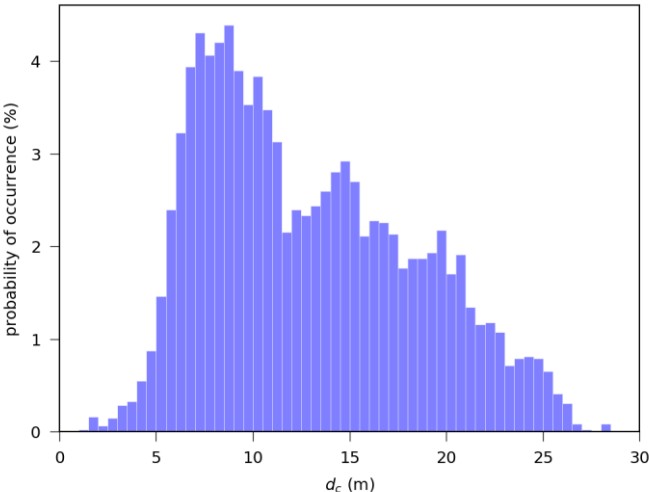

Figure 3: Histogram of the probability of occurrence of $d_c$ globally (using bins of 0.5 m).

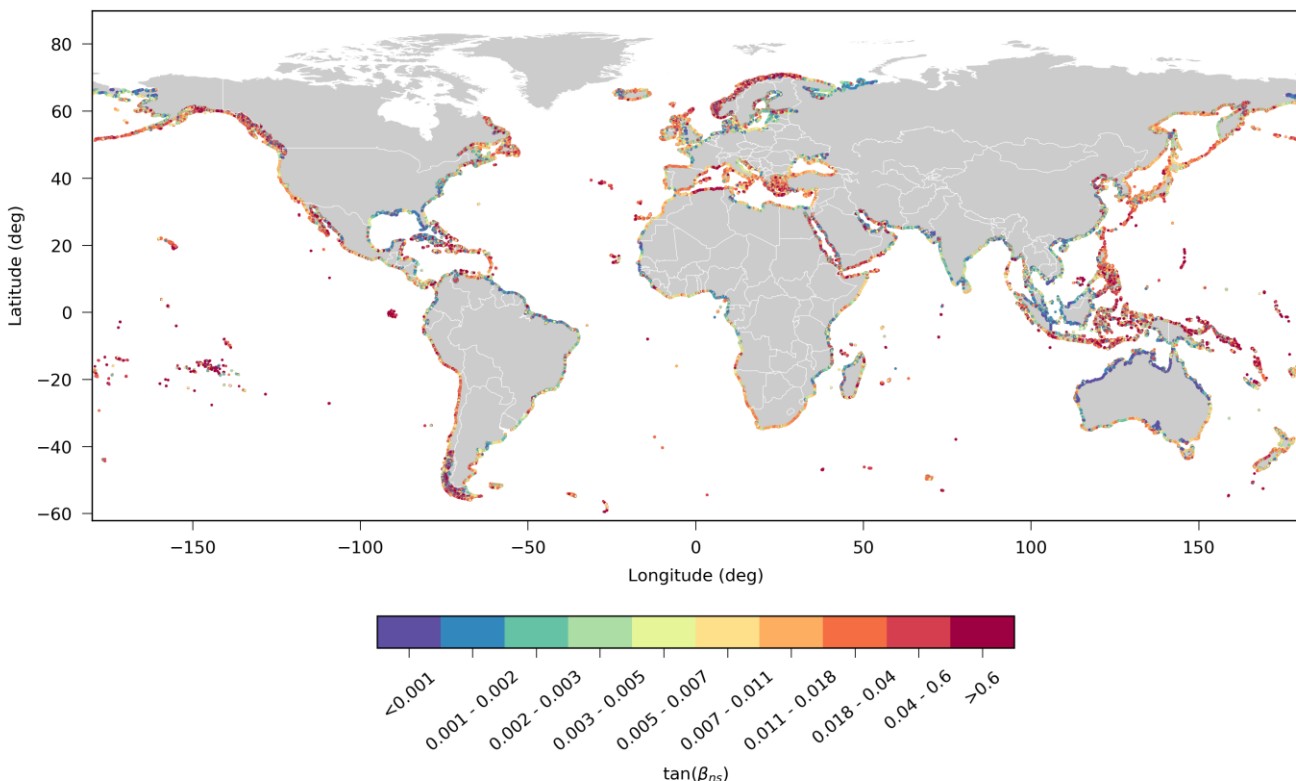

Figure 4: Global map of nearshore slopes. Red colours indicate steeper slopes while blue colours milder slopes. Note that in the colour scale the slopes have been grouped in non-equidistant increments in order to highlight the spatial differences.

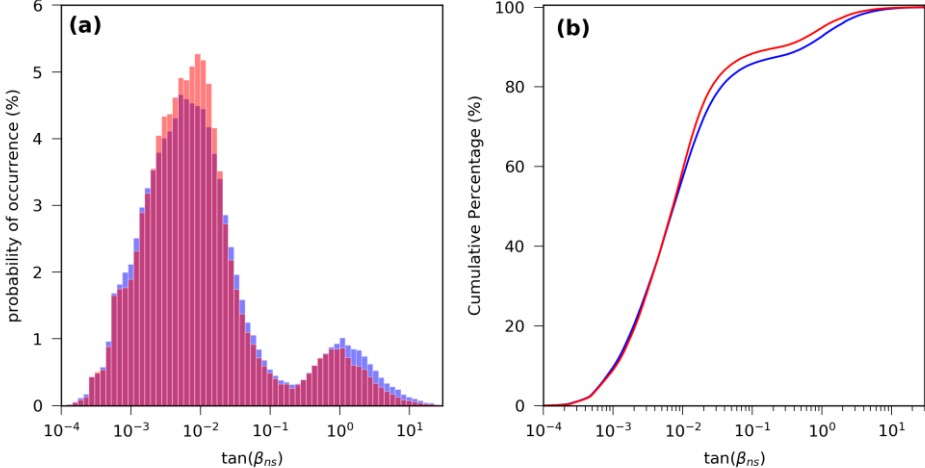

Figure 5: (a) Histogram of the probability of occurrence of nearshore slopes. (b) Cumulative probability distribution of nearshore slopes. The graphs have been plotted for all transects (blue) and only for sandy transects (red). Note that the x axis is plotted in log scale.



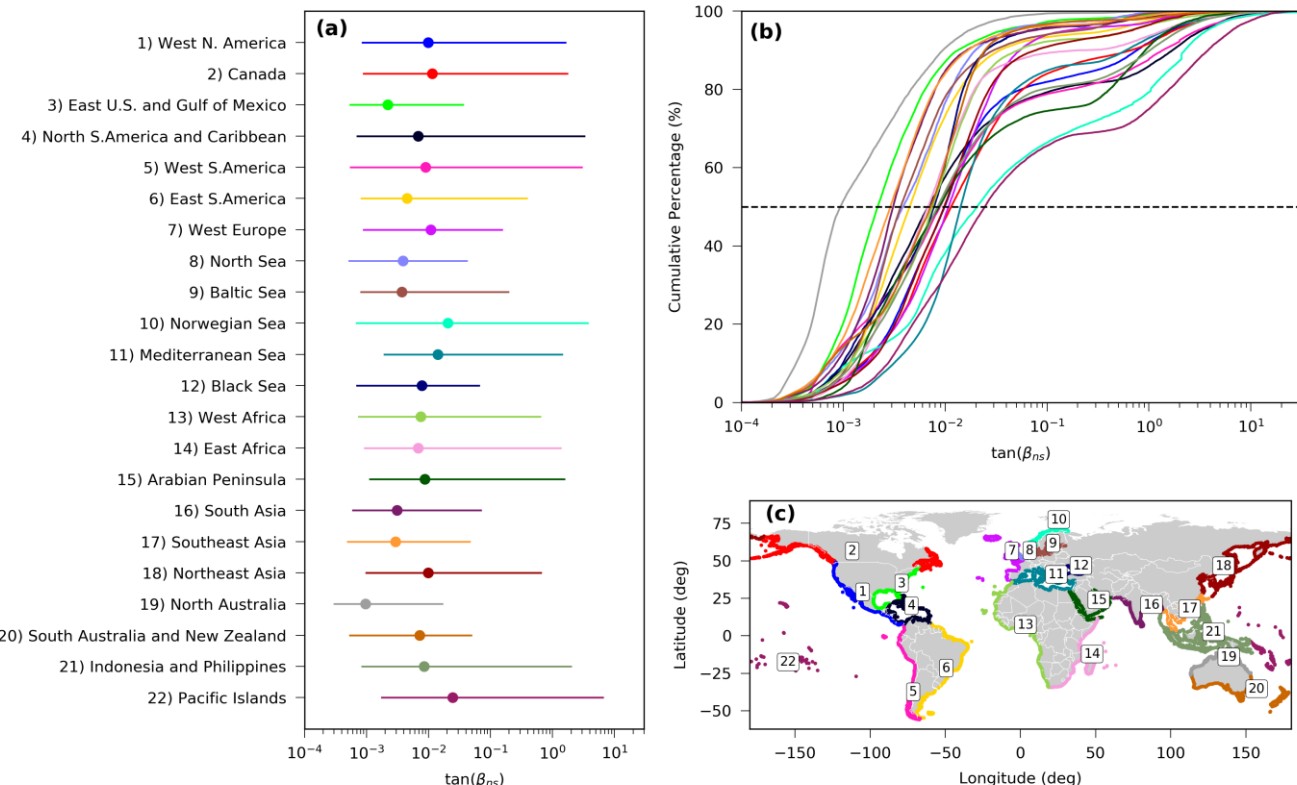

Figure 6: (a) Nearshore slopes statistics per region. Dots indicate the median, while lines the range between the $5^{th}$ and $95^{th}$ percentiles. (b) Cumulative probability distribution of nearshore slopes per region. Dashed line indicates the 50% line. (c) Global map with the defined regions with their respective colour and id. Note that the sub-figures a and b use the colour scheme indicated by sub-figure c. The x-axis of sub-figures a and b are plotted at a log scale.



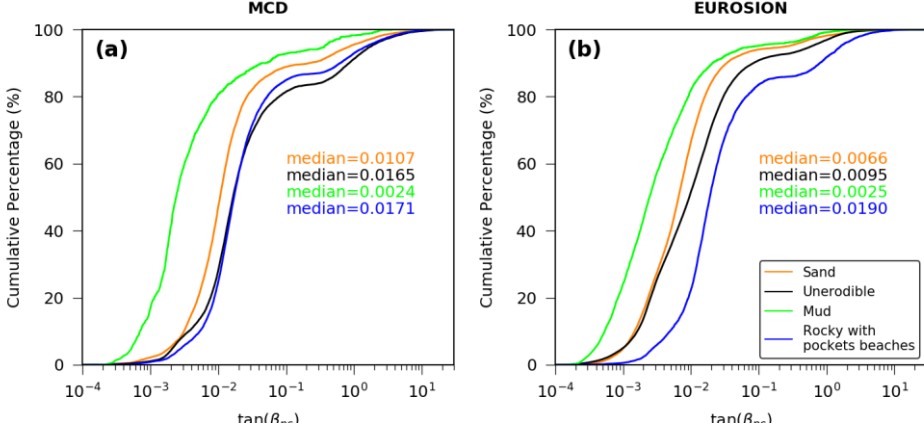

Figure 7: Cumulative probability distribution of nearshore slopes per coastal type for the (a) MCD dataset and (b) EUROSION dataset. The median values per coastal types are plotted with their respective colours as indicated in the legend. Note that the x axis is plotted in log scale.



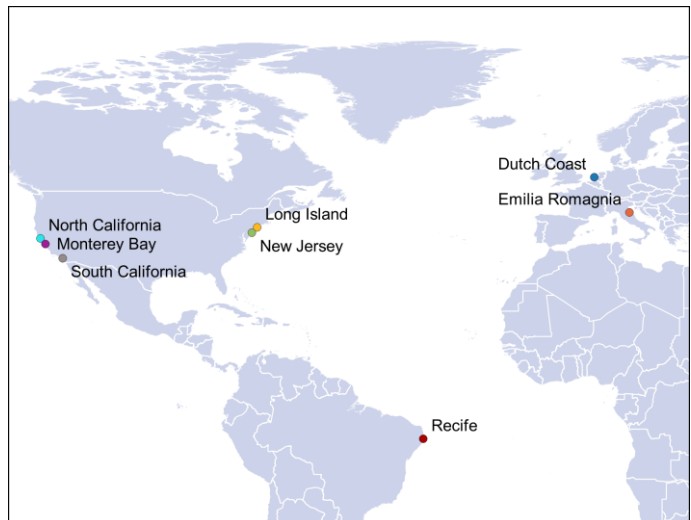

Figure 8: Field sites used for validation of the estimated global nearshore slopes.



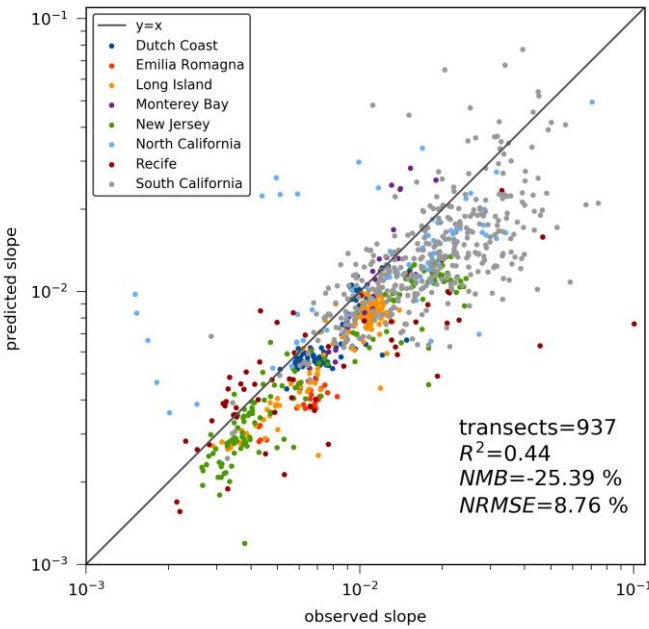

Figure 9: Linear regression between globally predicted nearshore slopes and nearshore slopes observed from local data for all study sites. Note that both axis are in log scale.

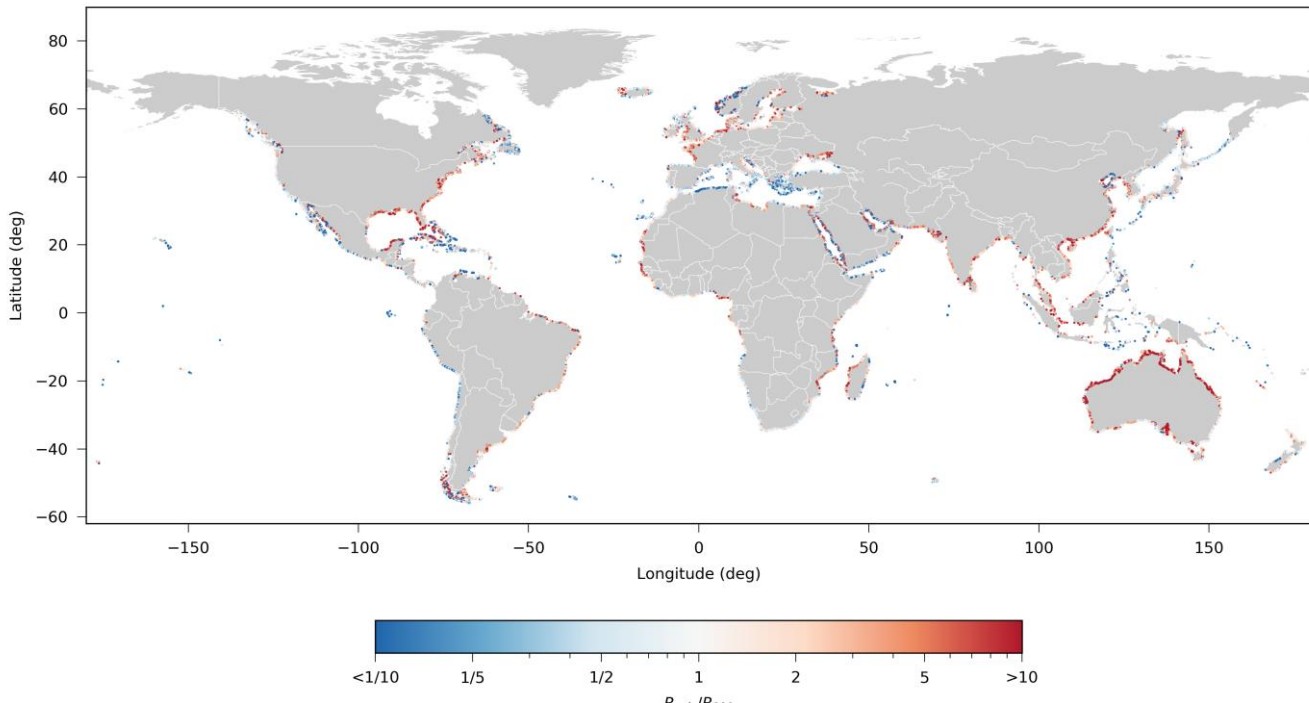

Figure 10: Global map of $\frac{R_{calc}}{R_{100}}$ where red colours indicate locations where the $tan(\beta_{sf}) = 1:100$ assumption underestimates the coastal recession while blue colours locations where it overestimates it. Note that the colourbar uses a log scale.



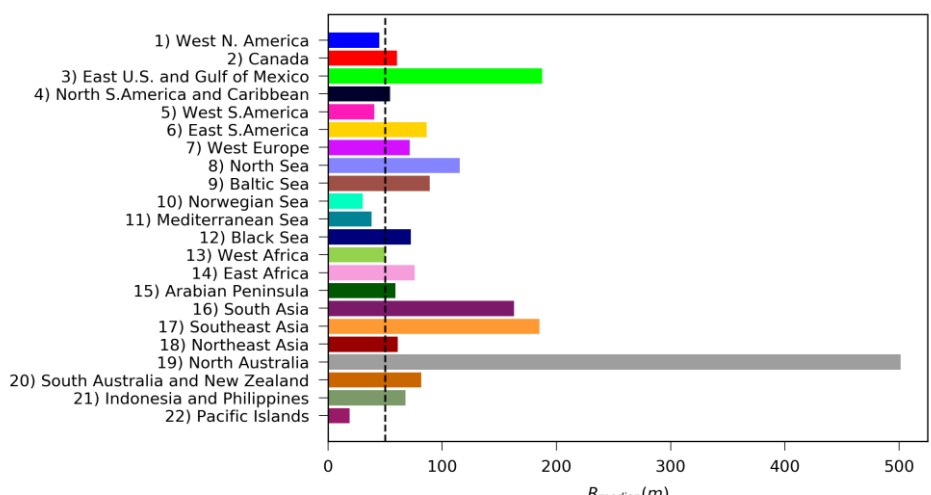

Figure 11: Median coastal recession in meters, calculated per region for the sandy transects assuming 0.5 m of SLR across the global coastline. The vertical dashed line indicates the recession of 50 m, which would be the result if a 1/100 nearshore slope was assumed.

