# Peer review of "Global distribution of nearshore slopes with implications for coastal retreat"

_Earth System Science Data, 2019_

## Referee Comment (RC1) · Anonymous Referee #1 · 25 Jun 2019

The manuscript by Athanasiou et al. introduces a new global dataset of nearshore slopes with the aim to address an important gap of global-scale coastal analyses. The manuscript is very well structured and the methods are presented in a comprehensive and understandable manner. From a methodological point of view, the work that the authors have carried out is very meticulous. Data processing has been conducted with high attention to detail and limitations of the work are explicitly acknowledged. Importantly, based on the description that is provided, the results are reproducible and the methods can be employed by other researchers when improved data become available. I recommend that the manuscript is published. I have included below a small number of comments that the authors should address in order to clarify some methodological decisions and accordingly discuss/revise some small parts of the manuscript.

I hope that the authors find the comments useful for improving the manuscript.

- pg. 1 line28: the coast also comprises different land use/cover types, which also define its response to these events

- pg. 3 line 23: Based on my experience, the choice of the coastline is crucial for the analysis. According to the smoothing method described, a different coastline dataset (which could diverge considerably from the selected one) would influence the results (see also discussion by Lichter et al., 2010, in the context of exposure analysis). I believe that this is a point that the authors need to discuss (briefly) in the manuscript (there is mention of this in the limitations section, but not really addressing the point)

- pg. 4 lines 1-4: To my understanding, the smoothing process can considerably affect the calculation of the slope and generally produce much lower values. I am surprised that the authors do not discuss this point in the limitations. Also, doesn't this process essentially "remove" all cliffs? Finally, isn't this the reason why the calculated global slopes are milder than those observed? (see lines 5-8, pg. 8).

- pg. 4 line 14: I am a little confused with the interval (25m) chosen for the equidistant points along the transects, considering the horizontal resolution of the employed datasets. Please justify.

- pg. 4 line16: what do "significant human interventions" include?

- pg. 5 line 18: how many of those not calculated were sandy beaches?

- pg. 8 line 20: is it the wave climate affecting slope or vice versa? Also, swell waves often occur in coasts with mild slopes. I believe that the explanation here is rather weak.

- A general point: to my experience, the calculation of slope in a global application can be affected by the type of projection. Also, re-projecting datasets to different projections can be a complex (or even problematic) task in the context of global applications. I think that the authors would need to provide some additional information (also in the

supplementary material if they prefer) regarding those choices and state what type of projections they employed and how slope/length were calculated for the entire world.

---

## Referee Comment (RC2) · Anonymous Referee #2 · 17 Jul 2019

Is the article appropriate to support the publication of a data set? Yes, indeed. It describes the dataset, how data were compiled and/or produced and it also provides some examples of application.

Is the data set significant – unique, useful, and complete? The described and online provided dataset is significant, useful, worldwide distributed and it will help to fill a frequent commonly found gap in coastal data.

Is the data set itself of high quality? Due to its worldwide distribution, the data set can be considered of good quality. They have been validated against local data and bias and error measurements are provided. Is the data set publication, as submitted, of high quality? The data set publication is adequate to the importance and quality of the data provided. The length to describe the data is appropriated, and the publication itself has

the right structure and cover the required aspects to be useful for readers and potential users of the data.

In what follows some specific comments are given.

[1] Pag 2, line 8. The used definition of the depth-of-closure is not the standard one. The depth-of-closure defined by equation 1 is NOT the offshore limit where sediment transport is zero. The standard definition is "the depth seaward of which there is no significant change in bottom elevation".

[2] Pag 2, line 22. As it is written this sentence seems to indicate that in the presented study the depth of closure is going to be calculated in a different way of those mentioned in the previous sentence (empirical formulas using wave parameters).

[3] Page 5, line 1. According to the authors, dc is calculated using the standard definition of wave height associated with a probability of occurrence of 12 hours/year but applied to a long (34 years) time series. Although used time series is long enough to be representative, it is not infrequent (Udo & Takeda reference provided by authors is an example) to assume that when dc is going to be used in long-term processes characterization, this average wave climate can be substituted by other characteristic of extreme conditions (e.g. by using a wave height associated with a given return period). Which should be the expected impact of this approach in the dataset?

[4] Page 5, line 2. dc is calculated using the Vousdoukas et al (2018b) wave data. However, there is no information on how good is the data set. In fact, in the original reference there is not too much mention to the calibration of hindcast against measured wave data. Since this parameter is used to define the profile length to compute the nearshore slope, which is the potential impact of the accuracy (or lack of) on the obtained values.

[5] Page 5, lines 5-6. Authors select dc in an opposite manner as it has been traditionally defined, i.e. it is usually calculated as the most landward position whereas authors'

method results in the most seaward location.

[6] Page 6, line 23. This is not a validation of dc computations. Authors are comparing their estimation with another estimation. Since in both cases they are using the same equation (no dc data are available in the comparison), they are essentially comparing used wave data, i.e. they are checking whether Vousdoukas and WIS wave data are comparable (at statistical terms). Due to this, sentence in line 28 is not fully true. Differences can simply be explained by the difference in wave statistics at a given location among both wave data sources.

[7] Due to the above-mentioned questions regarding dc and its implications for calculating nearshore slope, it would be interesting which is the impact of changing dc in different % (shallower and deeper) in the estimated nearshore slope. This will help to assess the robustness of estimated values as well as the real impact of the selection of the dc-value on the data set due to the base data (shallow water bathymetry) accuracy and resolution.

[8] Page 8, lines 20-23. The generally steeper slope along the West Americas coast is explained by using the role of swell as well as tectonics. Which should be the most dominant factor when dealing with the inner shelf slope? Is also sediment size any factor to be considered?

[9] Section 5.2 apparently deals with possible improvements in estimation in SLR-induced retreat. However, it is only a comparison between estimations using the provided dataset against the use of a common and single value worldwide (1/100). In this sense, it is just an improvement when comparing in those conditions. However, if these estimations are compared with many of the existing ones using local data, the improvement is not so evident.

[10] Limitations is an important aspect to be considered. In this case, validation of the dataset needs to be improved by increasing local datasets to compare along world coastlines under different conditions and characteristics.

---

## Author Comment (AC1) · 21 Aug 2019

We would like to thank both the reviewers for the careful and thorough reading of this manuscript and for the thoughtful comments and constructive suggestions they provided. They undoubtedly helped to improve the quality of this manuscript. In this response we are addressing the comments and feedback from the reviewers. We have numbered the reviewers comments (R1.1, R1.2 etc. for Reviewer 1 and R2.1, R2.2 etc. for Reviewer 2) in order to facilitate referencing to each comment. The pages and line numbers in our responses refer to the manuscript with the changes.

Anonymous Referee #1 (R1) The manuscript by Athanasiou et al. introduces a new global dataset of nearshore slopes with the aim to address an important gap of globalscale coastal analyses. The manuscript is very well structured and the methods are presented in a comprehensive and understandable manner. From a methodological point of view, the work that the authors have carried out is very meticulous. Data processing has been conducted with high attention to detail and limitations of the work are explicitly acknowledged. Importantly, based on the description that is provided, the results are reproducible and the methods can be employed by other researchers when improved data become available. I recommend that the manuscript is published. I have included below a small number of comments that the authors should address in order to clarify some methodological decisions and accordingly discuss/revise some small parts of the manuscript. I hope that the authors find the comments useful for improving the manuscript.

R1.1 - pg. 1 line28: the coast also comprises different land use/cover types, which also define its response to these events

As suggested by Reviewer 1 we included this missing part in our description:

pg.1 line28: "As the coastline comprises various different landforms (e.g. sandy coasts, rocky cliffs etc.) and land uses (e.g. heavily urbanized or natural), the response to these hazards can vary significantly both spatially and temporally."

R1.2 - pg. 3 line 23: Based on my experience, the choice of the coastline is crucial for the analysis. According to the smoothing method described, a different coastline dataset (which could diverge considerably from the selected one) would influence the results (see also discussion by Lichter et al., 2010, in the context of exposure analysis). I believe that this is a point that the authors need to discuss (briefly) in the manuscript (there is mention of this in the limitations section, but not really addressing the point)

We did not explicitly test the sensitivity of the elevation profiles to the choice of coastline, but we considered other coastlines in the beginning of the study. The issue we encountered was that other datasets either lacked the required resolution (NOOA coastline) or did not have a global coverage and consistency (various national datasets).

That is why we decided to use the OSM coastline. Nevertheless, we agree that this is an important issue that we now mention in the discussion:

Pg.10 line 28: "Although other available coastline datasets exist, either they have a coarser resolution, therefore not allowing for a correct representation of the separation between land and water, or do not have a global coverage. The importance of the coastline and DEM data has been previously highlighted with respect to statistics of exposure information (Lichter et al., 2011). We expect that the choice of the coastline can have an effect in the elevation profiles, especially in areas with a high tidal range."

R1.3 - pg. 4 lines 1-4: To my understanding, the smoothing process can considerably affect the calculation of the slope and generally produce much lower values. I am surprised that the authors do not discuss this point in the limitations. Also, doesn't this process essentially "remove" all cliffs? Finally, isn't this the reason why the calculated global slopes are milder than those observed? (see lines 5-8, pg. 8).

We understand the thinking of Reviewer 1 on this topic. With respect to the cliffs we believe that the calculated slopes may indeed be milder than the actual ones, but they can still be rather steep. This can be verified by the global distribution of slopes as well (Figure 5), where a high number of quite steep slopes exist (steeper than 1/2), related to the large length of coastline with cliff features. Since our study focused on non-cliffed coastlines we did not test this further. In relation to the second part of the question, we believe that, since the smoothing is applied in areas close to the shoreline, this will not have a substantial effect on the areas close to the depth of closure, and hence will not affect the estimates of slopes in areas where the slopes are not very steep. For example even in the Dutch coast where no smoothing was needed, there is a negative bias (Table 2). We believe that the negative bias is connected with general differences of the accuracy of the GEBCO bathymetry. Nevertheless, to address the reviewer comment we have now included in the discussion section the following sentences:

Pg.11 line 5: "The smoothing procedure that was applied to the bathymetric and topographic data may have affected the nearshore slope values, especially at cliffed coastlines, where the computed slopes may be milder than the reality. Nevertheless, cliffed transects retained quite steep values as can be seen by the second peak in the global slope distribution (Figure 5)."

R1.4 - pg. 4 line 14: I am a little confused with the interval (25m) chosen for the equidistant points along the transects, considering the horizontal resolution of the employed datasets. Please justify.

The choice of a 25m interval results from a number of sensitivity tests to ensure that 1) the cross-shore resolution is captured well and 2) the amount of data saved is manageable.

R1.5 - pg. 4 line16: what do "significant human interventions" include?

We are sorry for the confusion. Here we were mainly focusing on ports, when talking about "significant human interventions". We have now clarified this:

Pg.4 line 23: "...exclude transects located along coasts covered with ice, river inlets, or major ports, using the global transects of Luijendijk et al. (2018)."

R1.6 - pg. 5 line 18: how many of those not calculated were sandy beaches?

Using the same sandy beach detection dataset (Luijendijk et al. 2018), the estimated number of sandy transects for which the slope could not be calculated is ∼33,000 of the 160,000 (∼20%). We have now included this in the manusript:

Pg.5 line 28: "The transects of which the nearshore slope could not be calculated were about 160,000 of the total 780,000. Almost 20% out of these 160,000 transects were identified as sandy."

R1.7 - pg. 8 line 20: is it the wave climate affecting slope or vice versa? Also, swell waves often occur in coasts with mild slopes. I believe that the explanation here is rather weak.

In the long term (geological time scales) the waves can shape a coast and in the shorter term the coastal slopes affect the wave transformation. There are various processes that can also have a major effect in shaping the coastal profile like the sediment supply from rivers, the tectonics, the presence of submarine canyons etc. We have now tried to reformulate our discussion section with respect to these:

Pg.9 Line 13: "At sedimentary coasts the upper nearshore profile is mainly influenced by waves while the lower one is influenced by tectonics, sediment supply or major storm events."

R1.8 - A general point: to my experience, the calculation of slope in a global application can be affected by the type of projection. Also, re-projecting datasets to different projections can be a complex (or even problematic) task in the context of global applications. I think that the authors would need to provide some additional information (also in the supplementary material if they prefer) regarding those choices and state what type of projections they employed and how slope/length were calculated for the entire world.

Actually, for our study we projected the OSM coastline to the local UTM projection and then created the transects. In this way we ensured that the spacing between the transects is consistent globally and that all the transects have the same (real) length. We have now added two sentences to the Methods sections supporting our choice:

Pg.4 line 13: "To ensure a globally consistent spacing and length of the transects, the coastline was re-projected to the local UTM projection, before creating the transects."

Pg.4 line 20: "For each transect the longitude and latitude of the extraction points were used to sample from the global merged elevation grid, and create a continuous elevation profile." 

Anonymous Referee #2 (R2) Is the article appropriate to support the publication of a data set? Yes, indeed. It describes the dataset, how data were compiled and/or produced and it also provides some examples of application. Is the data set significant – unique, useful, and complete? The described and online provided dataset is significant, useful, worldwide distributed and it will help to fill a frequent commonly found gap in coastal data. Is the data set itself of high quality? Due to its worldwide distribution, the data set can be considered of good quality. They have been validated against local data and bias and error measurements are provided. Is the data set publication, as submitted, of high quality? The data set publication is adequate to the importance and quality of the data provided. The length to describe the data is appropriated, and the publication itself has the right structure and cover the required aspects to be useful for readers and potential users of the data. In what follows some specific comments are given.

R2.1 Page 2, line 8. The used definition of the depth-of-closure is not the standard one. The depth-of-closure defined by equation 1 is NOT the offshore limit where sediment transport is zero. The standard definition is "the depth seaward of which there is no significant change in bottom elevation".

We appreciate the comment by Reviewer 2. We have now changed the definition to:

Pg.2 line 8: "...between the depth of closure d_c (i.e. the depth seaward of which there is no significant change in bottom elevation) and the shoreline (MSL)."

R2.2 Page 2, line 22. As it is written this sentence seems to indicate that in the presented study the depth of closure is going to be calculated in a different way of those mentioned in the previous sentence (empirical formulas using wave parameters).

We can understand that this sentence might send the wrong message to the reader. We have now changed it to:

Pg.2 line 21: "The most common approach to estimate d_c is using empirical formulae (Hallermeier, 1978; Birkemeier, 1985; Nicholls et al., 1998) that relate d_c to wave parameters. This has been employed in several studies both at the global scale (Hinkel

et al., 2013) and regional scale (Brutsché et al., 2016; Toimil et al., 2017)."

R2.3 Page 5, line 1. According to the authors, dc is calculated using the standard definition of wave height associated with a probability of occurrence of 12 hours/year but applied to a long (34 years) time series. Although used time series is long enough to be representative, it is not infrequent (Udo & Takeda reference provided by authors is an example) to assume that when dc is going to be used in long-term processes characterization, this average wave climate can be substituted by other characteristic of extreme conditions (e.g. by using a wave height associated with a given return period). Which should be the expected impact of this approach in the dataset?

We think Reviewer 2 might have misunderstood our approach on this matter. We used the 34 year reanalysis of wave heights and periods to calculate the wave height exceeded only 12 hour in these 34 years. This means that we take the wave height with an exceedance probability of 0.137% in these 34 years, not the average of the yearly values. Nevertheless, as we understand that the current version might be confusing, we have now tried to make it more clear in the manuscript:

Pg.5 Line 13: "Following Nicholls et al. (1998), here we calculate $H_{(e,t)}$ as the wave height that is not exceeded more than 12 hours over the full 34 years' time series. This approach is expected to lead to larger $d_c$ values than taking the yearly statistics, as larger waves will be included in the time series."

R2.4 Page 5, line 2. dc is calculated using the Vousdoukas et al (2018b) wave data. However, there is no information on how good is the data set. In fact, in the original reference there is not too much mention to the calibration of hindcast against measured wave data. Since this parameter is used to define the profile length to compute the nearshore slope, which is the potential impact of the accuracy (or lack of) on the obtained values.

The reference we used previously did not include the validation of the wave model. We have now replaced the reference with the one, in which the detailed information of the
validation can be found in the appendix:

Pg.5 Line 9: "Here we applied Eq. (1) using the full time series of significant wave heights H_s and peak wave periods T_p from the global reanalysis covering the period between 1980 and 2014 presented by Vousdoukas et al. (2017) . Waves were simulated using the third generation spectral wave model WW3. The model has been extensively validated and detailed information can be found in the reference provided."

R2.5 Page 5, lines 5-6. Authors select dc in an opposite manner as it has been traditionally defined, i.e. it is usually calculated as the most landward position whereas authors' method results in the most seaward location.

We agree with Reviewer 2 that this is not the traditional way to define the depth of closure. Nevertheless, this would make a difference mainly in systems with pronounced sandbar systems, which cannot be explicitly captured with the present resolution. We have tested the sensitivity to this option and the differences in the calculated slopes were insignificant. We did these for the 8 validation sites that we use in our study and found that the error statistics had minor differences between the two methodologies (in the order of 2%).

R2.6 Page 6, line 23. This is not a validation of dc computations. Authors are comparing their estimation with another estimation. Since in both cases they are using the same equation (no dc data are available in the comparison), they are essentially comparing used wave data, i.e. they are checking whether Vousdoukas and WIS wave data are comparable (at statistical terms). Due to this, sentence in line 28 is not fully true. Differences can simply be explained by the difference in wave statistics at a given location among both wave data sources.

In order to do an explicit validation of dc values the only option would be to compare with dc values derived locally by numerous surveys to identify the location of no bed level change. This approach is quite challenging as long term local datasets are not always available. That is why we decided to at least validate our results with respect

to previous studies which used the same formulations. Nevertheless, we agree that this is not a direct validation of the dc values. Therefore we have now used the word comparison in place of validation, as follows:

Pg.7 Line 4: "A thorough validation of the estimated depths of closure would demand various local bathymetrical surveys with good spatial resolution and sufficiently long temporal resolution and extent (i.e. 34 years of data). This kind of information is not readily available. Therefore, here we compare our results with"

R2.7 Due to the above-mentioned questions regarding dc and its implications for calculating nearshore slope, it would be interesting which is the impact of changing dc in different % (shallower and deeper) in the estimated nearshore slope. This will help to assess the robustness of estimated values as well as the real impact of the selection of the dc-value on the data set due to the base data (shallow water bathymetry) accuracy and resolution.

We have re-calculated the nearshore slopes globally by adjusting the local depth of closure with a percentage of the originally estimated one. This adjustment was chosen to be -20%, -10%, +10% and +20% of the original value. We have now added the sensitivity test results to the manuscript and supplement:

Pg.12 Line 5: "Another important matter is the sensitivity of the estimated nearshore slopes to the depth of closure values used in this study. In order to test the robustness of our dataset, and how different wave statistics or depth of closure formulae could affect our estimations, we varied the previously calculated $d\_c$ values (Sect. 2.3). Changes of -20%, -10%, 10% and 20% of the originally estimated $d\_c$ were tested. The normalized differences of the estimated nearshore slopes had moderate sensitivity to the depth of closure variation (Fig. S12). When the depth of closure was increased and decreased by 10%, at almost 70% of the transects, the nearshore slope changed by the same percentage. On the other hand, with an increase and decrease of the $d\_c$ by 20%, the nearshore slope difference was in the order of 20% at about 75% of

the transects. The number of transects with changes up to 20% was higher when only the sandy transects were considered. In general, the largest increase in the nearshore slope was observed when d_c was decreased. The spatial distribution of the relative nearshore slope changes showed that in most case the sensitivity was low. The most sensitivities areas to changes in d_c were areas with steep slopes (e.g. fjord areas in Norway, Chile and others) (Fig. S13)."

Figure S 12 (Fig. 1 in this response) : Histogram of normalized nearshore slope difference for all the transects (blue) and sandy transects (red) for four changes in the depth of closure (-20%, -10%, 10% and 20%). The data have been grouped in bars in increments of 10%.

Figure S 13 (Fig. 2 in this response): Global maps of normalized nearshore slope difference for all the transects for four changes in the depth of closure (-20%, -10%, 10% and 20% from top to bottom).

R2.8 Page 8, lines 20-23. The generally steeper slope along the West Americas coast is explained by using the role of swell as well as tectonics. Which should be the most dominant factor when dealing with the inner shelf slope? Is also sediment size any factor to be considered?

It is quite challenging to thoroughly address the question of which is the dominant factor across the entire West Americas coast. Nevertheless we have now tried to include all the factors that could play a role and reformulate our discussion:

Pg.9 Line 9: "Sediment grain size can play an important role as well on the shape and slope of the nearshore profile (Dean, 1991)."

Pg.9 Line 12: "The local slope values are ultimately the results of the combination of all the physical processes mentioned previously. At sedimentary coasts the upper nearshore profile is mainly influenced by waves while the lower one is influenced by tectonics, sediment supply or major storm events."

R2.9 Section 5.2 apparently deals with possible improvements in estimation in SLR induced retreat. However, it is only a comparison between estimations using the provided dataset against the use of a common and single value worldwide (1/100). In this sense, it is just an improvement when comparing in those conditions. However, if these estimations are compared with many of the existing ones using local data, the improvement is not so evident.

As the reviewer points out, the improvement we highlight here is indeed that of using our dataset against the assumption of globally uniform nearshore slope (e.g. Hinkel et al. 2013). We agree with the reviewer that if local observations are available they would provide more accurate coastal retreat estimates locally. But for regional, continental or global studies that seek to quantify trends and hotspots, our dataset can provide an important improvement by incorporating the spatial variability of nearshore slopes versus a uniform slope assumption.In the manuscript we highlight the purpose of this comparison in the introduction and in the discussion sections (Pg. 2 Lines 5-7 and Pg. 9 Lines 13-17)

R2.10 Limitations is an important aspect to be considered. In this case, validation of the dataset needs to be improved by increasing local datasets to compare along world coastlines under different conditions and characteristics.

We agree with the comment of the reviewer. For this work, we have made use of most of the authors' contacts globally to get as much data as we could. The challenge is that local bathymetric data are sometime not publicly available. We expect that since our publication and dataset are open access, researchers in possession of local data could further validate our estimates against their measurements. We have additionally added a small part in the Discussion section that highlights the need for more data:

Pg.11 Line 25: "More long term coastal profile datasets, including coastal sites in diverse coastal environments, are needed to ensure a thorough validation of our dataset. The wealth of long term satellite data combined with the constantly evolving remote

sensing techniques can be potentially an avenue to consider in the future."

[Figure]

[Figure]

**Fig. 1.** Caption in comment (R2.7)

[Figure]

**Fig. 2.** Caption in comment (R2.7)